# Optimal and Generalizable Multimodal Representation Learning Framework through Adaptive Graph Construction

## Abstract

Multimodal contrastive learning train neural networks by levergaing data from heterogenous sources such as images and text. Yet, current multimodal learning architectures cannot generalize to an arbitrary number of modalities, need to be hand-constructed and are often not robust to missing modalities. We propose AutoBIND, a novel contrastive learning framework that can learn representations from an arbitrary number of modalities. AutoBIND uses a graph-based approach to automatically select the most correlated modalities and uses a contrastive loss to learn the representations. AutoBIND is robust to missing modalities as it can dynamically update the graph structure during training. Based on the proposed framework, each modality maps to a shared embedding space and that the correlation between two modalities can be used as a measure of similarity between the two modalities. Therefore, the graph structure improves dynamically during training, purely as a result of the minimization of the contrastive loss. We evaluate AutoBIND on a wide variety of datasets, including tasks such as Alzhiemer's disease detection and house price prediction, and across a broad range of data modalities: 3D images, 2D images, and tables. We also show that AutoBIND outperforms previous methods on these tasks, highlighting the generalizablility of the approach.

## 1 Introduction

The human brain possesses a remarkable capacity to learn and integrate information from diverse sensory modalities in a dynamic manner. This is similar to the challenge of multimodal contrastive learning. Just as the brain synthesizes inputs from various senses like images, sound, text and numbers to form a cohesive understanding of the world, multimodal contrastive learning aims to train neural networks by leveraging data from different sources.

Among existing multimodal learning approaches, several methods have made significant strides. Hager et al. (2023) proposes a method for handling tabular and image data through the combination of self-supervised learning and contrastive learning. However, it exhibits limitations when confronted with more than two modalities, and furthermore, its approach exclusively supports single modality outputs, thus constraining its versatility.

ImageBIND Girdhar et al. (2023), specializes in binding multiple modalities but within a fixed number. This technique utilizes image data to harmonize diverse modalities. However, its weakness emerges when confronted with missing image data, rendering it less robust in such scenarios.

Huang (2023) presents the latest framework for multimodal contrastive learning of medical images and tabular data, applying the techniques to Alzheimer's disease prediction. With carefully crafted contrastive neural topologies, it claims over 83.8% prediction accuracy, 10% increase from the state of the art solutions.

Unfortunately, all these existing multimodal contrastive learning approaches, although promising, fall short in achieving generalization across an arbitrary number of modalities. They demand meticulous manual construction and often prove vulnerable when dealing with absent modalities. These methods are inherently shaped by the specific knowledge and assumptions associated with the

modalities they address, making them less adaptable to scenarios requiring universality. To the best of our knowledge, no prior work in multimodal contrastive learning has proposed solutions to tackle the challenge of learning with an arbitrary number of modalities in a universally applicable manner.

In this work, we present **AutoBIND**, a versatile and universally applicable framework for contrastive learning with an arbitrary number of modalities. To effectively learn and refine the representations of each modality, AutoBIND employs contrastive loss that highlights the nuanced relationships between the data sources. A distinguishing feature of AutoBIND is its dynamic graph structure adaptation during training. This adaptive graph mechanism is a pivotal solution for handling the absence of certain modalities, an issue that often plagues traditional multimodal learning techniques. Just as the human brain adapts to the absence of certain sensory inputs by amplifying the significance of others, AutoBIND dynamically updates the graph's topology to effectively accommodate and learn from the available modalities.

Drawing inspiration from the successes of ImageBIND on image data, AutoBIND capitalizes on the emergent binding properties observed between indirectly connected modalities during the training process. These binding properties lead to the phenomenon where similar representations gravitate towards each other while dissimilar ones become distinct, effectively organizing the modality representations into meaningful clusters. Leveraging this insight, AutoBIND employs a graph-based updating strategy, using a minimum spanning tree approach to pinpoint the most correlated modalities. Through this strategic graph adaptation, the method orchestrates a dynamic, data-driven evolution that enhances the quality of learned representations and bolsters the model's ability to handle missing modalities effectively.

Our contributions are as follows. **(1)** For the first time to our knowledge, we propose and tackle the problem of constrative learning with arbitrary number of modalites. We formulate the problem as a graph based edge-weight minimization problem, where the nodes are the modality embeddings and edge weights are the similarity between modalities. **(2)** We propose **adaptable** solutions with different graph construction methods, including a fully connected graph and a minimum spanning tree. We show that the graph structure improves dynamically during training, purely as a result of the minimization of the contrastive loss. **(3)** We implement a dynamic graph update mechanism for **automatic organization** of the graph, which allows the model to handle missing modalities by preserving the available data's integrity and adjusting the graph representation accordingly, which makes the framework **robust** as a result. Furthermore, using the optimal learned framework from the minimum spanning tree, we further improve the performance of multimodal learning through modality pruning. **(4)** We evaluate AutoBIND on a wide variety of datasets, including tasks such as Alzhiemer's disease detection and house price prediction, and across a broad range of data modalities: 2D and 3D images and tables. We show that AutoBIND outperforms previous methods on these tasks, highlighting the generalizablility and **modal-agnostic** capability of the approach.

## 2 PROBLEM SETUP

In our setup, we consider aribitary $n$ modalities, where each modality $i$ has a set of instances $X_i$ and an encoder function $f_i$ that learns representations $Z_i$ for each modality. Our goal is to learn a shared embedding space where representations of similar instances across different modalities are brought closer together, while representations of dissimilar instances are pushed apart. This is achieved by optimizing an objective function that involves both positive and negative pairs of instances.

This loss function aims to maximize the similarity between positive pairs and minimize the similarity between negative pairs of instances. Mathematically, the objective to be minimized can be expressed as:

$$\mathcal{L} = \sum_{i=1}^{n} \sum_{j \neq i} \sum_{x_i \in X_i} \sum_{x'_j \in X_j} - \log \left( \frac{\exp(\text{Sim}(z_i, z'_j))}{\exp(\text{Sim}(z_i, z'_j)) + \sum_{x''_j \in X_j} \exp(\text{Dissim}(z_i, z''_j))} \right) \quad (1)$$

Where the outer summation goes over all modalities $i$ and $j$ where $i \neq j$, and the inner summation goes over all instances $x_i$ in modality $i$ and all instances $x'_j$ and $x''_j$ in modality $j$.

In multimodal contrastive learning, the primary constraint to be satisfied is that the representations $Z_i$ are learned in such a way that they align well across modalities. This alignment ensures that instances from different modalities with similar semantics are represented closely in the shared embedding space. The optimization process aims to find encoder functions $f_i$ that minimize the contrastive loss while satisfying the constraint of meaningful alignment of representations across modalities.

## 2.1 Graph Representations

We represent the problem of multimodal constrastive learning as an undirected graph $G = (V, E)$, where $V$ is the set of nodes, each corresponding to a modality $i$, and $E$ is the set of edges, where an edge $(i, j)$ represents the correlation between modality $i$ and modality $j$. For each edge $(i, j) \in E$, we can use a similarity function $\text{Sim}(Z_i, Z_j)$ to express the correlation between modalities $i$ and $j$ in the shared embedding space.

We utilize the cosine similarity as our edge weight function to quantify the correlation between different modalities. The cosine similarity, often employed to measure the similarity between vectors, provides a suitable metric to gauge the relationship between modalities' representations in the shared embedding space. Given representations $Z_i$ and $Z_j$ for modalities $i$ and $j$, the cosine similarity $\text{Sim}(Z_i, Z_j)$ is defined as:

$$w_{ij} = \text{Sim}(Z_i, Z_j) = \frac{Z_i \cdot Z_j}{\|Z_i\|\|Z_j\|}$$

Where $Z_i \cdot Z_j$ denotes the dot product of the representations and $\|Z_i\|$ and $\|Z_j\|$ represent their respective Euclidean norms. This similarity score captures the directional agreement between modalities, providing a value between -1 (dissimilar) and 1 (similar). By utilizing the cosine similarity as the edge weight function, we effectively establish the foundation for modeling correlations and dependencies among modalities within the graph structure, ultimately shaping the optimization process to reflect these relationships in the learning objectives.

The distance between nodes on the graph is indicative of the correlation between modalities. For two modalities $i$ and $j$, the distance $d_{ij}$ between them can be directly related to the similarity $\text{Sim}(Z_i, Z_j)$ using a decreasing function:

$$d_{ij} \propto \frac{1}{\text{Sim}(Z_i, Z_j)}$$

.

Hence, smaller distances indicate higher correlation between modalities.

To summarise, the proposed graph representation reflects the organization and interactions of modalities, and optimization techniques can position nodes strategically to minimize contrastive loss, aligning with the objective of accentuating correlations and reducing disparities. In the following paragraphs, we demonstrate that graphs not only provide a visual understanding of modalities' relations but also facilitate efficient problem-solving through graph-based algorithms, making them an attractive solution for modeling the intricate interplay between modalities and guiding the optimization process effectively.

## 2.2 Graph Optimization

The optimization objective involves minimizing the contrastive loss, which can be expressed using the similarity and dissimilarity functions as follows:

$$\mathcal{L}(Z_i, Z_j) = -\log\left(\frac{\exp(\text{Sim}(Z_i, Z_j))}{\exp(\text{Sim}(Z_i, Z_j)) + \sum_{k \neq i} \exp(\text{Dissim}(Z_i, Z_k))}\right) \quad (2)$$

Now, let's consider two sets of modalities $i$ (correlated modalities) and $j$ (uncorrelated modalities). The graph-based argument states that arranging correlated modalities ($i$) together in the graph leads to a lower overall loss than mixing them with uncorrelated modalities ($j$):

$$\sum_{m \in i} \mathcal{L}(Z_m, Z'_m) + \sum_{n \in j} \mathcal{L}(Z_n, Z'_n) < \sum_{p \in i \cup j} \mathcal{L}(Z_p, Z'_p) \qquad (3)$$

Where $Z_m$ and $Z'_m$ are representations of correlated modalities $m$, and $Z_n$ and $Z'_n$ are representations of uncorrelated modalities $n$. $Z_p$ and $Z'_p$ are representations of modalities in the combined set $i \cup j$.

At the end of the graph optimization process, where nodes representing correlated modalities are strategically positioned closer together and uncorrelated modalities are separated, this arrangement directly influences the behavior of the original contrastive loss $\mathcal{L}$ to be minimized.

## 3 PROPOSED METHODS

### 3.1 GRAPH CONSTRUCTION

We consider two different approaches to constructing the graph: fully connected graph (FCG) and minimum spanning tree (MST). The fully connected graph is the simplest approach, where each modality is connected to every other modality. The minimum spanning tree is a tree-based approach that selects the most correlated modalities and connects them together.

**Fully Connected Graph.** A fully connected graph (FCG) is selected as the initial representation, denoted by $G_{\text{full}} = (V, E_{\text{full}})$, where each modality $i$ corresponds to a node in $V$, and the edge set $E_{\text{full}}$ includes all possible edges $(i, j)$ between modalities. This choice is made to comprehensively capture modalities' intricate correlations within the contrastive learning framework. By including all edges, we ensure that every modality participates in the correlation calculations, allowing the model to account for potential dependencies across the entire multimodal spectrum.

**Minimum Spanning Tree.** The adoption of a minimum spanning tree (MST), achieved through Kruskal's algorithm, is motivated by the desire to distill essential correlations while simplifying the graph structure. The MST, denoted as $G_{\text{MST}} = (V, E_{\text{MST}})$, is a subgraph of the FC graph that retains the fundamental correlations while eliminating excess edges. This simplification is crucial for interpretation and computational efficiency. The choice of the MST aligns with the understanding that not all correlations are equally essential, and by preserving the core relationships, we can reduce noise and redundancy, leading to a more focused and interpretable graph. This distilled structure guides the next step of the process, node pruning. We remove the nodes with lowest sum of the edge weights, as they are the least correlated with other modalities. This pruning process is repeated until a desired number of modalities is reached.

### 3.2 GRAPH UPDATE

In each iteration ($batch\_num$), the algorithm follows these steps:

---

**Algorithm 1:** Multimodal Minimum Spanning Tree (MMST) Algorithm

---

**Data:** List of modalities
**Result:** Minimum Spanning Tree $G_{\text{MST}}$

1 Initialize an empty list of edges $E_{\text{empty}}$;
2 **for** *each pair of modalities $i$ and $j$* **do**
3    Calculate the correlation factor between $i$ and $j$ and store it in $E_{\text{empty}}$;
4 **end**
5 Sort $E_{\text{empty}}$ in non-decreasing order of correlation weights;
6 Form an empty graph $G_{\text{MST}}$ representing the minimum spanning tree;
7 Utilize a disjoint set data structure to track connected components;
8 **foreach** *sorted edge $(u, v)$* **do**
9    **if** *adding edge $(u, v)$ to $G_{MST}$ does not create a cycle* **then**
10      Incorporate edge $(u, v)$ into $G_{\text{MST}}$;
11      Merge disjoint sets of nodes $u$ and $v$;
12      Train the contrastive model on modality pair $(u, v)$ using get_embedding($u$, batch_num)
       and get_embedding($v$, batch_num);
13    **end**
14 **end**
15 Visualize the resulting minimum spanning tree $G_{\text{MST}}$, capturing core correlations while
   ensuring a simplified graph structure for interpretability and efficiency purposes;

---

Based on the derived edge weights, we prune the nodes with lowest sum of the edge weights, calculated by

$$\text{node\_weight}(i) = \sum_{j \neq i} w_{ij}$$

This pruning process has two benefits. First, it reduces the number of modalities, which reduces inference time for a target task. Second, it improves the performance of the model by removing the least correlated modalities, which can be detrimental to the learning process. This pruning process is repeated until a desired number of modalities is reached.

After using this approach, we finetune the model on the resulting graph $G_{\text{MST}}$.

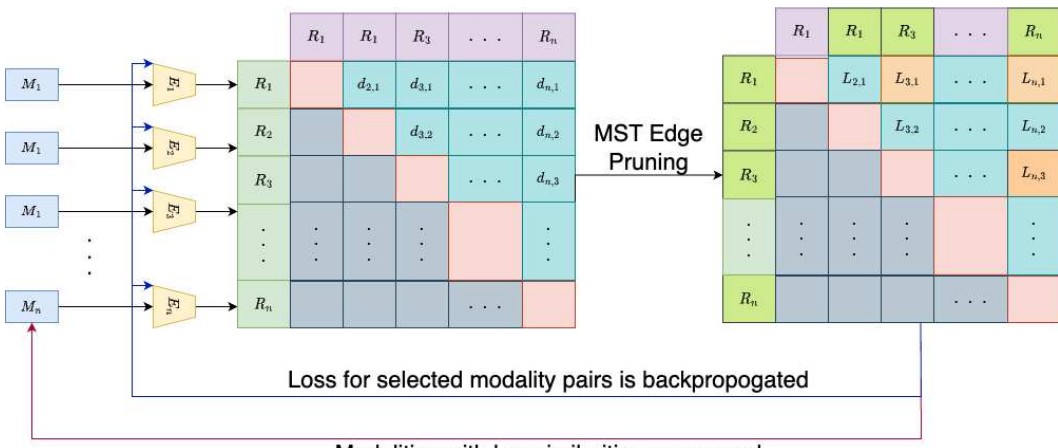

Figure 1: Overview of the AutoBIND Process: Illustration depicting the various stages and steps involved in the AutoBIND framework. The process encompasses multimodal embedding, graph construction, and node pruning, resulting in enhanced performance across different datasets.

### 3.3 MODEL ARCHITECTURE

In our contrastive learning experiments, we employed distinct encoder architectures to effectively capture the inherent characteristics of different modalities. These encoders play a critical role in transforming the raw data from each modality into meaningful and semantically rich representations.

**Image Encoder.** For processing image data, we utilized a ResNet-34 architecture, a popular choice in computer vision tasks. The ResNet-34 architecture is equipped with a Multi-Layer Perceptron (MLP) Projection head, which projects the intermediate features into a compact 128-dimensional vector space. This projection enhances the network's ability to distill relevant information from the images.

**Text Encoder.** To process textual data, we employed DistilBERT, a variant of the BERT model that strikes a balance between efficiency and performance. DistilBERT is adept at capturing contextual information from text, making it a suitable choice for our text modality. Similar to the image encoder, we attached an MLP Projection head to the DistilBERT model to create 128-dimensional embeddings.

**Tabular Encoder.** Tabular data demands a unique approach due to its structured nature. We utilized a Tabular Attention mechanism from Huang (2023) for encoding tabular data. This attention mechanism allows the encoder to focus on different parts of the input table while generating representations. As with the other encoders, an MLP Projection head was employed to map the tabular representations to a 128-dimensional space. The same encoder is used to create an embedding for the labels in the chosen dataset.

## 4 EXPERIMENTS

### 4.1 EVALUATION TASKS

**Classification Tasks.** Our evaluation methodology unfolds through a series of strategic steps. Firstly, we encode each modality while encompassing every conceivable label class. Subsequently, the embeddings originating from each modality are thoughtfully concatenated. Leveraging the potency of cosine similarity, we pinpoint the class exhibiting the highest similarity score. This classification accuracy stands as our primary metric, embodying the core of our evaluation process. To provide a more comprehensive understanding of our model's behavior, we also delve into secondary metrics such as precision.

**Regression Tasks.** We begin by encoding individual intervals within a designated range of values pertinent to the label. Subsequently, the embeddings originating from each modality are adeptly merged. Employing cosine similarity once again, we determine the interval that resonates most closely. The mean squared error (MSE) of the predicted label emerges as our primary metric for regression evaluation.

### 4.2 RESULTS

We assess the versatility of our model on the ADNI database Jack Jr et al. (2008) and House Prices Dataset Ahmed & Moustafa (2016), encompassing a diverse range of modalities such as images and tablular values, rendering them conducive to the realm of multimodal contrastive learning.

#### 4.2.1 ALZHIEMER'S DISEASE DETECTION

We utilize the Alzheimer's Disease Neuroimaging Initiative (ADNI) dataset, a comprehensive collection of multimodal data encompassing subjects with a spectrum of cognitive states, including normal cognition, mild cognitive impairment (MCI), and Alzheimer's disease (AD). The ADNI dataset comprises diverse modalities, including tabular data and medical images. Furthermore, the ADNI dataset contains missing tabular values in the tabular data, making it an ideal candidate for evaluating our model's robustness in the face of missing modalities.

For the image data, we evaluate our model performance on both 2D and 3D images. The 2D images are 3-plane-localizer MRI, while the 3D images are fused MRI-PET images.

For the tabular data extraction from the ADNI dataset, we utilize the ADNIMerge tool. The extracted tabular data consists of diverse columns, which we categorize into various groups to capture different aspects of patients' information. These categories encompass biomarkers like APOE4 and pTau, empirical cognitive assessments including MMSE and RAVLT Arevalo-Rodriguez et al. (2015), volumetric measurements like hippocampus size and brain size, and medical history details such as the baseline diagnosis.

| MODEL | MODALITY | ACCURACY | PRECISION | RECALL | M-M | APT | RES | T-A |
|---|---|---|---|---|---|---|---|---|
| 2D RESNET | 2D MRI | $0.799 \pm 0.000$ | $0.770 \pm 0.124$ | $0.799 \pm 0.000$ | - | - | - | - |
| 3D CNN | 3D MRI-PET | $0.745 \pm 0.064$ | $0.594 \pm 0.082$ | $0.799 \pm 0.000$ | - | - | - | - |
| MEDBIND | 2D MRI | $0.838 \pm 0.023$ | $NA \pm NA$ | $0.799 \pm 0.000$ | ✓ | - | - | ✓ |
| AUTOBIND MST | MULTI | $\mathbf{0.916 \pm 0.014}$ | $\mathbf{0.936 \pm 0.012}$ | $\mathbf{0.933 \pm 0.014}$ | ✓ | ✓ | ✓ | ✓ |
| AUTOBIND FCG | MULTI | $0.848 \pm 0.028$ | $0.877 \pm 0.020$ | $0.864 \pm 0.028$ | ✓ | ✓ | - | ✓ |

Table 1: Performance for AutoBIND on ADNI Datasets
(M-M: multimodal, APT: adaptability, RES: resilience to missing data, T-A: task agnostic)

We compare our model with three baselines: a 2D ResNet baseline Sun et al. (2021) and a 3D CNN baseline Song et al. (2021) and MedBIND Huang (2023). The 2D ResNet baseline is a ResNet-34 architecture trained on the 2D images. The 3D CNN baseline is a 3D CNN architecture trained on the 3D images. We further compare the impact of each individual modality on the model's performance by doing unimodal prediction. We also compare the performance of different graph construction methods, including FCG and MST.

Furthermore, we plot the finetuned graph representation of the ADNI dataset using 2D images. The graph is constructed using the MST method. The nodes represent different modalities, and the edges represent the correlations between them.

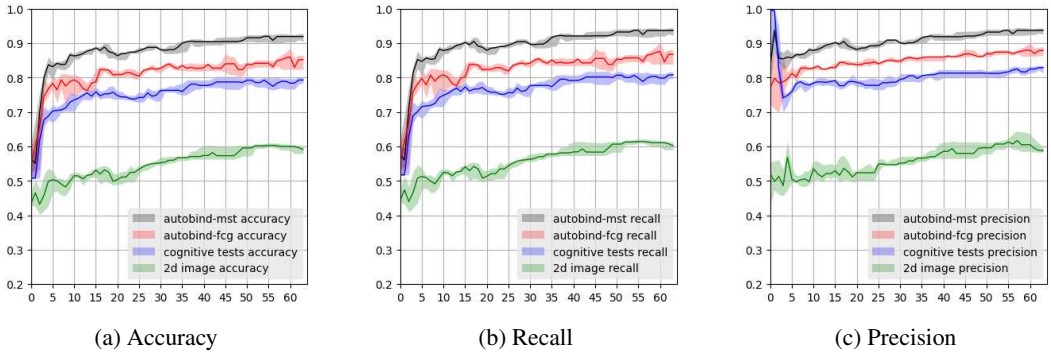

(a) Accuracy        (b) Recall        (c) Precision

Figure 2: Performance of AutoBIND MST vs. AutoBIND FCG vs Unimodals.

From Table 1, we can see that AutoBIND MST outperforms the baseline models and MedBIND in terms of accuracy, precision and recall. This shows that the MST graph construction method and node pruning is more effective than the FC graph construction method. Furthermore, AutoBIND FCG also outperforms the baseline models and MedBIND in terms of accuracy, precision and recall. This illustrates the importance of multimodal contrastive learning in improving performance.

Compared to existing works, both AutoBIND graph construction methods are task agnostic, meaning that they can be applied to any task in any domain. Furthermore, they are also adaptable to different datasets and encoders, whereas existing models are task-specific and rely on static frameworks for multimodal learning. Finally, our models are also resilient to missing data, as the tabular data in the ADNI dataset contains missing values.

We can visualize the graph generated as a result of MST graph construction in Figure 3. The nodes represent different modalities, and the edges represent the correlations between them. This enables interpretability, as we can see the correlations between different modalities. For example, we can see that the medical history and 2D MRI modality is highly correlated with the presence of Alzheimer's disease. Node pruning is a crucial step in the AutoBIND process. In this case, we only keep the following nodes: images, label, medical history, cognitive tests. These are the modalities most correlated to the presence of Alzheimer's, and thus the most important for the model to learn. This reduces the number of modalities from 7 to 4, which reduces inference time for disease prediction.

Furthermore, these insights gleaned from the AutoBIND-generated graph underscore the capacity of this approach not only to enhance predictive accuracy but also to provide valuable interpretative cues for understanding the complex interplay of data modalities in the diagnosis of Alzheimer's

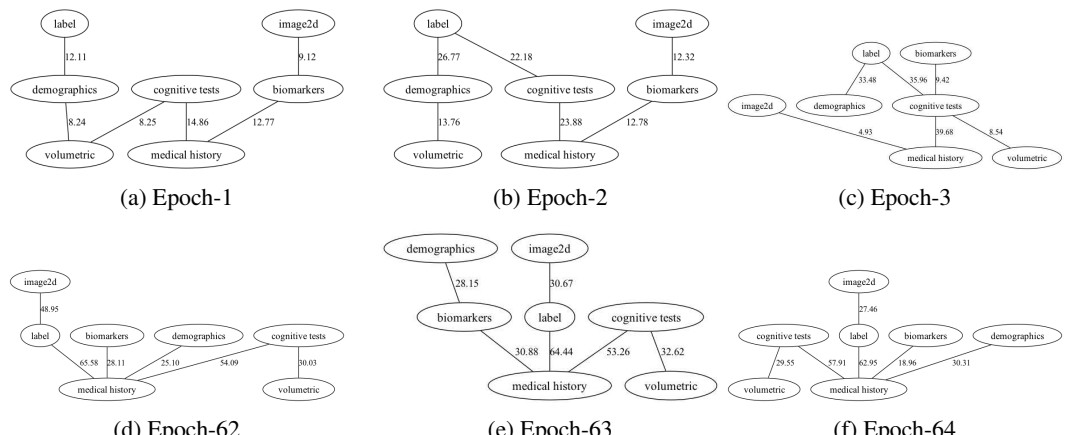

Figure 3: Graph Visualization of AutoBIND w/ 2D MRI

| MODEL | REPRESENTATION | MSE | ADAPTABILITY | TASK AGNOSTIC |
|---|---|---|---|---|
| LINEAR REGRESSION | - | 1.573E5 ± 9.263E4 | - | - |
| IMAGEBIND | STATIC | 1.279E5 ± 6.075E4 | - | - |
| AUTOBIND | FCG | 1.212E5 ± 7.012E4 | ✓ | ✓ |
| AUTOBIND | MST | **8.307E4 ± 3.043E4** | ✓ | ✓ |

Table 2: Performance for AutoBIND on House Prices Dataset

disease. Such interpretability is invaluable for both clinical practitioners and researchers in the field, facilitating a deeper comprehension of the disease's diagnostic landscape.

### 4.2.2 HOUSE PRICES PREDICTION

In addition to the ADNI dataset, we introduce the House prices dataset Ahmed & Moustafa (2016), tailored for house price estimation. This dataset incorporates both visual and textual information about houses, with each property represented by four images: bedroom, bathroom, kitchen, and a frontal house image. The dataset contains 2140 images and a tabular text file containing metadata, such as bedroom count, bathroom count, area, zipcode, and price.

We compare each modality's performance on the model's overall performance by doing unimodal prediction. We also compare the performance of other models, including ImageBIND and linear regression, which only uses the tabular data.

Furthermore, we plot the finetuned graph representation of the House Prices dataset. This graph is constructed the same way as the ADNI experiment.

We can see that AutoBIND outperforms the other methods in terms of mean squared error, showing the generalizability of our approach to different modalities and domains. Furthermore, the MST approach outperforms the other approaches, showing the effectiveness of the approach in capturing the core correlations while reducing complexity.

Similarly to the ADNI experient, we plot the graph created by the MST in Figure 4. We can see that the tabular data is highly correlated with the labels. This is because the tabular data contains information about the house, such as the number of bedrooms, bathrooms, and the area. These variables are highly correlated with the price of the house, and therefore, the tabular data is highly correlated with the labels. We can also see that the frontal image is highly correlated with the tabular data. This is due to the fact that the frontal image captures the visual appearance of the house, including its architectural features and aesthetics. These visual attributes are inherently linked to the tabular data, as the number of rooms and the house's size directly influence its visual appearance. Therefore, the strong correlation between the frontal image and tabular data highlights how AutoBIND can effectively capture the complementary relationship between different data modalities, enriching the model's ability to predict house prices accurately.

| MODALITY | MSE |
|---|---|
| FRONTAL IMAGE | $1.399\text{E}5 \pm 4.609\text{E}4$ |
| BATHROOM IMAGE | $4.993\text{E}5 \pm 4.236\text{E}5$ |
| BEDROOM IMAGE | $4.803\text{E}5 \pm 2.607\text{E}5$ |
| KITCHEN IMAGE DATA | $4.858\text{E}5 \pm 2.178\text{E}5$ |
| TABULAR DATA | $1.214\text{E}5 \pm 6.258\text{E}4$ |
| **MULTIMODAL (ALL DATA)** | $\mathbf{8.307\text{E}4 \pm 3.043\text{E}4}$ |

Table 3: Performance for Unimodal Prediction using MST

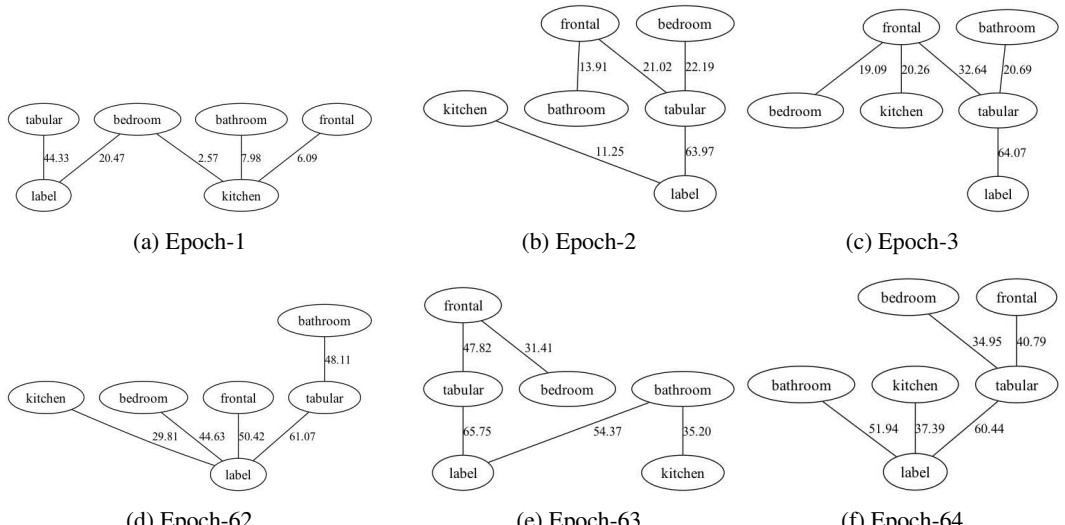

(a) Epoch-1       (b) Epoch-2       (c) Epoch-3

(d) Epoch-62       (e) Epoch-63       (f) Epoch-64

Figure 4: **Graph Visualization of AutoBIND w/ House Dataset**

## 5 DISCUSSION

In culmination, our endeavor has yielded a novel approach to multimodal contrastive learning, unveiling a methodology that surmounts numerous challenges to deliver noteworthy outcomes across a spectrum of datasets. Demonstrating the prowess of our approach, we have showcased its superior performance across various datasets. Specifically, it surpasses existing methods such as ImageBIND on the ADNI dataset, thereby underlining its resilience in the face of missing modalities. Moreover, its competitive edge on the House Prices dataset substantiates its adaptability and generalizability in diverse domains beyond the confines of medical data. Our method further attains superiority on the HAIM-MIMIC dataset, a true testament to its capacity to transcend conventional image and tabular modalities, validating its versatility in accommodating more intricate data sources.

Notably, the orchestrated graph structure, a cornerstone of our methodology, converges to an optimal framework for multimodal learning. This adaptively evolving structure dynamically adapts to the available modalities, effectively enhancing learning and accommodating variable data configurations.

As we chart a course for future exploration, we propose the investigation of alternative graph structures that could further refine our model's performance. Our curiosity extends to uncharted modalities such as audio and video, aiming to harness the power of our approach in realms beyond text, images, and tables.

However, we acknowledge the limitations of our approach. A paucity of datasets with correlated modalities poses a challenge to robust validation. Moreover, the $O(n^2)$ complexity of the graph construction algorithm restrains scalability to an extensive array of modalities. Addressing these constraints will be pivotal in enhancing the applicability of our method in real-world scenarios with a multitude of modalities.

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

## A APPENDIX

### A.1 TRAINING DETAILS

In pursuit of effective model convergence and optimal performance, our training approach encompasses carefully chosen parameters. For the ADNI dataset, we adopt a batch size of 32, while the House Prices Dataset is trained with a batch size of 64, and the HAIM-MIMIC dataset employs a batch size of 16. Training extends over 100 epochs for each dataset, integrating an early stopping mechanism that intervenes when the validation loss remains stagnant for 10 consecutive epochs, thereby averting overfitting. The Adam optimizer guides our weight updates, employing a learning rate of 0.0001 to facilitate efficient convergence. The contrastive loss function is modulated by a temperature parameter of 0.1, which governs the scale of the feature space. Notably, the training process is facilitated by the computational power of a V100 GPU.

### A.2 DATA PREPROCESSING DETAILS

For both the ADNI and House prices datasets, we perform comprehensive data preprocessing to ensure that the data is ready for use in our experiments. In this section, we provide in-depth information about the specific preprocessing steps undertaken for each dataset.

### A.2.1 ADNI DATASET PREPROCESSING

Categorical features within the ADNI dataset are handled using one-hot encoding. This encoding technique transforms categorical variables into a binary representation that can be effectively

integrated into our models. For label encoding, we adopt a spectrum-based approach, assigning numerical values to labels: CN (cognitively normal) as 0, MCI (mild cognitive impairment) as 0.5, and AD (Alzheimer's disease) as 1.

Numerical feature normalization is accomplished through z-score normalization. This process standardizes numerical values to have a mean of zero and a standard deviation of one, ensuring that each feature contributes proportionally during model training.

For image data, we collect 2D and 3D T1 Weighted MRI images along with FDG-PET images. These images undergo a series of preprocessing steps, including Gradwarp for geometric distortion correction, B1 non-uniformity correction to rectify intensity discrepancies, and N3 processing to reduce intensity non-uniformity. Image fusion is then applied to combine MRI and PET images.

### A.2.2 HOUSE PRICES DATASET PREPROCESSING

In the case of the House prices dataset, categorical features are one-hot encoded, while numerical features undergo the same z-score normalization as the tabular data in the ADNI dataset.

