# OpenReview forum: "Optimal and Generalizable Multimodal Representation Learning Framework through Adaptive Graph Construction"
_ICLR.cc/2024/Conference — Submitted to ICLR 2024_

### Official Review · Reviewer_ZYST · 2023-10-31

**Soundness:** 2 fair
**Presentation:** 2 fair
**Contribution:** 2 fair
**Rating:** 5
**Confidence:** 4

**Summary:**

The paper introduces AutoBIND, a contrastive learning framework that can learn representations from an arbitrary number of modalities.
This approach leverages data from heterogeneous sources such as images and text and is able to deal with missing modalities as it can dynamically update the graph structure during training.
AutoBIND uses a graph-based approach to automatically select the most correlated modalities from fully connected graph/MST and a contrastive loss to learn the representations.
The method is evaluated on Alzheimer's disease detection and house price prediction with 3D images, 2D images, and table modalities and outperforms existing baselines.

**Strengths:**

The paper motivates well by tackling one of the most important problems in multimodality learning, which is dealing with a numerous number of modalities and missing modalities.
The author proposes a simple yet effective method for extending the ImageBind pairwise contrastive learning with graph learning to learn the relation between the modalities. Results demonstrate decent results in the two downstream datasets, showing the effectiveness and interpretability of the method.

**Weaknesses:**

(1) Although the paper claims to bind multimodalities, in practice, it deals with only three modalities (image, text, tabular) and their corresponding features (various columns in the data). Such a number of modalities is relatively common in multimodal learning, unlike current works that learn from up to five modalities that contain image, video, text, audio, and IMU.
(2) The modalities claimed in the current setting are more like different features in the same modalities (bedroom, bathroom, etc ...). Hence, the missing modality setting is more like missing features, which is a relatively explored problem.
(3) It will be better to include the
ImageBIND method as the baseline in Table 1, to confirm if it is the proposed method or the use of MM features that achieve performance improvement.
(4) Due to the loss design including all pairs of combination (or MST edges), it will be better to include a more rigorous run-time analysis (how the method scales in adding more modalities), computation resource used, and parameter used. It is very likely that the performance gain was from more parameters in the encoder.

Some typos
Alzhiemer's should be Alzheimer's in the title
4.2.1

**Questions:**

It is unclear to me what the message the author tries to convey of the epoch 123 vs. 62,63,64 in Figures 3 and 4.
For me, all of the graphs make sense in their own way, which is based on different interpretability of such a task.
Also, the change between 62,63,64 is also not clear in Figure 4. In 62 and 64, the label is dependent on various features. However, it is only correlated to two of them in 63. It will be better for the author to elaborate more on these findings.

---

> ### Author Response · Authors · 2023-11-19
> **Response to Reviewer ZYST**
>
> ### In relation to the weaknesses mentioned:
>
> While the paper only deals with image, text and tabular data, the tabular data is categorised into different independent values and these are encoded independently, which leads to many different sources of data.
>
> Current works that use image, video, text, audio, and IMU data learns from these modalities separately (e.g. {image, video}, {image, text}, {text, audio}), whereas our approach considers the scenario where lots of modalities are available in a single dataset.
>
> ### To answer the question:
>
> We include the graphs for epoch 1, 2, 3 vs 62, 63, 64 to show the convergence of the MST graph. As shown in Figure 3: Graph Visualization of AutoBIND w/ 2D MRI, the (Image) - (Label) - (Biomarker) component of the graph is consistent for all 3 of the final epochs. We note that some other modality connections may fluctuate (e.g. demographics) since they are not correlated with the presence of AD.

---

> > ### Comment · Reviewer_ZYST · 2023-12-05
> >
> > I have read the reviews and rebuttals.
> > The lack of ablation on the proposed method and the robustness against missing variables is missing as in weaknesses 1 and 3.
> > Similar points were also raised by the reviewer Ehq7.
> > I'll keep my rating.

---

### Official Review · Reviewer_pwEx · 2023-10-31

**Soundness:** 1 poor
**Presentation:** 1 poor
**Contribution:** 1 poor
**Rating:** 3
**Confidence:** 4

**Summary:**

This paper proposes a method name AutoBIND, which construct a graph on different modalities and dynamically update the graphs using a minimum spanning tree algorithm. This paper evaluates AutoBIND on a wide variety of datasets and show that AutoBIND outperforms previous methods.

**Strengths:**

1. This paper Introduces AutoBIND, a novel framework that efficiently handles multiple modalities and is robust to missing modalities through a dynamic graph-based approach. Utilizing graphs to learn multimodal representations is reasonable, as this structured data can effectively model the relationships between different modalities.
2. Extensive Experiments across various datasets demonstrate AutoBIND's effectiveness in multimodal contrastive learning tasks.

**Weaknesses:**

1. In section 2.1, the definition of \(d_{ij}\) appears problematic. How is the value determined when the cosine similarity \(Sim(Z_i, Z_j)\) is zero? Moreover, when \(Sim(Z_i, Z_j)\) approaches 0, \(d_{ij}\) can become extremely large (if positive) or extremely small (if negative). This definition seems to be flawed at a fundamental level.
2. What is the specific definitions of correlated and uncorrelated modalities?
3. The symbol system is a mess. i,j are modality indices in equation 1 but are sets in equation 3, making this paper is hard to understand.
4. Overall, the paper lacks a clear structure and appears to be written informally.

**Questions:**

NA

---

> ### Author Response · Authors · 2023-11-19
>
> # 1
>
> Note that d_ij in experimentation is not exactly 1/Sim(zi, zj). We introduce a large constant d_ij = 1/(Sim(zi, zj) + 10^9). Therefore, the similarity value is never negative.
>
> # 2
>
> We define correlated modalities as modalities that are capable of producing similar embeddings due to similarities in their data.
>
> We define uncorrelated modalities as modalities that are not capable of this.
>
> E.g. patient demographics is uncorrelated with the presence of Alzheimer's disease, while the MRI is highly correlated.

---

### Official Review · Reviewer_Ehq7 · 2023-10-31

**Soundness:** 2 fair
**Presentation:** 2 fair
**Contribution:** 1 poor
**Rating:** 3
**Confidence:** 5

**Summary:**

This paper proposes a method to learn multimodal representation using a graph representation. The target representation is per modality, i.e. each modality should yield a separate vector of a fixed size (=128). Each node in the graph represents a separate modality. A graph is constructed in an order to maximize the correlations between modalities iteratively. The method was evaluated on two multimodal datasets for prediction tasks.

**Strengths:**

A graph construction method was used and the resulting graph may be used to understand the internal correlation of data.
The proposed method is very simple to implement.

**Weaknesses:**

Technical descriptions are very unclear. Mathematical formalization is inconsistent and doesn't make sense. The algorithm description is incomplete. I think I can still guess what the paper tries to do, but not because the presentation is good, but because the method is extremely simple.

At high level, there is no proof or evaluation showing that this method is "optimal" or "generalizable". There is no proof that the proposed algorithm can optimize the global objective (eq 1).

Eq (1) is the global objective by choosing optimal {f_i}. z_i is instance Z_i should be the set. In the first equation in Sec 2.1 (doesn't have a numbering), the papers uses dot product/norms of set (Z). This must be z_i. But then w_ij should be defined in the whole set, so probably needs summation.

The next paragraph introduces d_ij, which is just 1/Sim(). There is no point of defining this term because this term never appears in the rest of the paper.

Eq (2) and (3) make no sense. First, i and j now represent sets, not instances. Eq (2) is only valid when the inputs are two different modalities. Eq (3) is passing same modality in each of the loss terms.

Algorithm 1 also uses in consistent and undefined terms like "correlation factor" "get_embedding". How do you train the contrastive model on modality pair? It's never explained, although I guess it's probably done using eq (2).

The evaluation is another (bigger) limitation of the current paper. This paper seeks to learn a "unimodal" representation that can be learned from multimodal sources. This representation takes one modality as an input and produces a representation of fixed size. This can be useful in many situations, like when there is only one modality present per instance at test time. This paper doesn't consider any situation like this in evaluation. Instead, it uses all the modalities and concatenates all the representations to get a *joint* representation. And of course, this leads to a better prediction performance than using a single modality, but this is pointless. Maybe Table 3 tried to evaluate the model when each modality is separately used. But Table 3 is never referred in the main body and there is not much explanation. More importantly, if Table 3 was indeed from the proposed method, this needs to be compared with some references (which learn multimodal representation like this paper). Currently this table delivers no information.

There is an arbitrary step of pruning nodes, but there's no ablation study showing its impact. So I assume that the authors needed to do this just to improve the performance. It was argued that the current method is robust against missing variables, but again there was no experiment designed to verify.

Overall, I think the current paper needs a lot more improvement before it can be published.

**Questions:**

Please see above and correct me if I was mistaken.

---

> ### Author Response · Authors · 2023-11-19
>
> ### In relation to the optimality and generalisability:
>
> The MST is optimal due to the following:
>
> Given two correlated modalities A and B:
>
> - Consider that A and B are not directly connected due to particular biased batch or underfitting in the encoders.
>
> - If A and B are never connected, then the correspondance between A and B will never be found. We cannot determine that modalities A and B are similar without comparing them.
>
> - However, if A and B are indirectly connected, then their similarity will increase, as they are bound to one embedding space. Let A connect to C connect to B, where C is some component of the graph. In the embedding space of the vector of C, modalities A and B will be closer than if they were in disjoint graphs. Therefore, in the next iteration, the similarity between A and B increases.
>
> - Hence, all data modalities should be connected whilst similar modalities are being grouped together.
>
> Now further consider two uncorrelated modalities A and D:
>
> - A and D will always be connected. However, they will be distant in the embedding space, since they have little/no similarity
> - Hence, their edge will be pruned and they will be far apart on the graph
> - Since their modalities have low correlation AND they are far apart on the graph, there is very little interference to the overall contrastive training process
>
> The MST maintains connectivity of all modalities, whilst increasing the correspondance between similar modalities. After multiple iterations, the graph converges and similar modalities are grouped together.
>
> As shown in Figure 3: Graph Visualization of AutoBIND w/ 2D MRI, the (Image) - (Label) - (Biomarker) component of the graph is consistent for all 3 of the final epochs.
>
> The converged MST highlights inherent similarities between two modalities across the entire dataset.
>
> ### In relation to node pruning:
>
> After convergence, we can do node pruning. Node pruning eliminates the noise in the loss caused by the unrelated modalities. Furthermore, it reduces training time as less encoders require back-propagation.
>
> ### In relation to eq (2)
>
> Eq (2) defines the contrastive loss between two modalities i and j. To elaborate on the training process as a whole:
>
> Once the MST is constructed, for each selected edge, we calculate the contrastive loss between the nodes it connects. Then, back-propagation is performed on the sum of the contrastive loss.

---

### Official Review · Reviewer_5eim · 2023-11-03

**Soundness:** 3 good
**Presentation:** 3 good
**Contribution:** 1 poor
**Rating:** 3
**Confidence:** 4

**Summary:**

This paper proposes AutoBIND, a new multimodal contrastive learning framework that can learn representations from any number of modalities without needing hand-crafted architectures. AutoBIND uses a graph-based approach to automatically select the most correlated modalities for contrastive learning. This allows it to be robust to missing modalities, as the graph structure is dynamically updated during training. Experiments across diverse datasets and modalities demonstrate AutoBIND's ability to generalize and superior performance compared with previous approaches.

**Strengths:**

1. The paper proposes a reasonable solution to an important problem.

2. The experimental results demonstrated satisfactory performance in a number of mixed-modality problem settings.

3. Visualization and interpretation of the graph structure formed during the learning are interesting.

**Weaknesses:**

1. The overall methodology is a simple application contrastive learning to multiple modalities. It is not clear why it outperforms previous approaches. Is it due to the additional modalities or due to one or several components in the proposed method?

2. The MMST is constructed in every epoch (and also pruned)? If so, will this process converge to a stable tree and how does the final tree depend on the initial encoding quality (which determine the initial tree).

3. The paper has the word "optimal" in the title, but there is no discussion or proof on the optimality.

4. Eq (3) is not explained clearly. Is it a theorem?

**Questions:**

See weaknesses.

---

> ### Author Response · Authors · 2023-11-19
>
> ### The overall methodology is a simple application contrastive learning to multiple modalities. It is not clear why it outperforms previous approaches. Is it due to the additional modalities or due to one or several components in the proposed method?
>
> The benefit comes from the addition of relevant modalites, as they reduce the variance component of the prediction error. The relevance of these modalities is determined through the graph based methodology.
>
> ### The paper has the word "optimal" in the title, but there is no discussion or proof on the optimality.
>
>
> The fundamental idea of contrastive learning leverages the inherent conceptual commonalities between different correlated modalities of data. These commonalities are mirrored by learnable correlations in the data itself. Therefore, by the transitive property, modalities with low contrastive loss (or high similarity) can be mapped to each other, and these mappings can be propagated only if the modalities are connected. This motivates the use of MST.
>
> The MST is optimal due to the following:
>
> Given two correlated modalities A and B:
>
> - Consider that A and B are not directly connected due to particular biased batch or underfitting in the encoders.
>
> - If A and B are never connected, then the correspondance between A and B will never be found. We cannot determine that modalities A and B are similar without comparing them.
>
> - However, if A and B are indirectly connected, then their similarity will increase, as they are bound to one embedding space. Let A connect to C connect to B, where C is some component of the graph. In the embedding space of the vector of C, modalities A and B will be closer than if they were in disjoint graphs. Therefore, in the next iteration, the similarity between A and B increases.
>
> - Hence, all data modalities should be connected whilst similar modalities are being grouped together.
>
> Now further consider two uncorrelated modalities A and D:
>
> - A and D will always be connected. However, they will be distant in the embedding space, since they have little/no similarity
> - Hence, their edge will be pruned and they will be far apart on the graph
> - Since their modalities have low correlation AND they are far apart on the graph, there is very little interference to the overall contrastive training process
>
> The MST maintains connectivity of all modalities, whilst increasing the correspondance between similar modalities. After multiple iterations, the graph converges and similar modalities are grouped together.
>
> As shown in Figure 3: Graph Visualization of AutoBIND w/ 2D MRI, the (Image) - (Label) - (Biomarker) component of the graph is consistent for all 3 of the final epochs.
>
> The converged MST highlights inherent similarities between two modalities across the entire dataset.
>
> After convergence, we can do node pruning. Node pruning eliminates the noise in the loss caused by the unrelated modalities. Furthermore, it reduces training time as less encoders require back-propagation.
>
> ### Eq (3) is not explained clearly. Is it a theorem?
>
> This is a claim that is part of the hypothesis. We say that the total loss, if we contrast uncorrelated modalities (Zn and Z'n) and correlated modalities (Zm, Z'm) separately (i.e. bring similar representations closer and move dissimilar representations apart), is less than if we contrast all modalities (Zp, Z'p) together, regardless of similarity.

---

### Meta-Review · Area_Chair_RneT · 2023-12-14

**Metareview:**

This paper presents a novel approach to learn representations from multiple modalities using a graph-based method. The paper has garnered attention for its proposed solution to a critical problem in multimodal learning and its experimental results across diverse datasets. However, this paper suffers from significant deficiencies. Key concerns include unclear technical descriptions and inconsistent mathematical formalizations. Additionally, the novelty and claimed optimality of the approach are questioned, with the authors' rebuttals failing to convincingly address these issues. The evaluation of the method is also found to be insufficient, lacking in rigorous testing and comparison with relevant methods. Concerns are also raised about the graph construction process, specifically the lack of clarity around node pruning and its impact, with no supporting ablation study. The authors are encouraged to thoroughly address these points in future submissions.

**Justification For Why Not Higher Score:**

The weaknesses in methodology, evaluation, and presentation collectively suggest that the paper requires substantial improvements before  publication. The authors are encouraged to address these issues comprehensively in future revisions.

**Justification For Why Not Lower Score:**

N/A

---

### Decision · Program_Chairs · 2024-01-16

Reject